# Economic and financial viability of a pig farm in central semi-tropical Mexico: 2022–2026 prospective

Francisco Ernesto Martínez-Castañeda[1☯], Nicolás Callejas-Juárez[2☯], Oscar Cuevas-Reyes[1‡], Nathaniel Alec Rogers-Montoya[3‡], Germán Gómez-Tenorio[4‡], María Elena Trujillo-Ortega[5‡], Claudia Giovanna Peñuelas-Rivas[6‡], Elein Hernandez[7‡]*

1 Instituto de Ciencias Agropecuarias y Rurales, Universidad Autónoma del Estado de México, Toluca, México, 2 Facultad de Zootecnia y Ecología, Universidad Autónoma del Estado de Chihuahua, Chihuahua, México, 3 Colegio de Postgraduados, Especialidad en Ganadería, Montecillos, México, 4 Centro Universitario Temascaltepec, Universidad Autónoma del Estado de México, Temascaltepec de González, México, 5 Facultad de Medicina Veterinaria y Zootecnia, Universidad Nacional Autónoma de México, Coyoacán, México, 6 Pipestone, FarmTeam Coordinator, Pipestone, Minnesota, United States of America, 7 Facultad de Estudios Superiores Cuautitlán, Department of Clinical Studies and Surgery, Universidad Nacional Autónoma de México, Coyoacán, México

☯ These authors contributed equally to this work.
‡ These authors also contributed equally to this work.
* elein_ht@comunidad.unam.mx

**Data Availability Statement:** All relevant data are within the manuscript and its Supporting Information files.

## Abstract

To estimate the economic and financial viability of a pig farm in central sub-tropical Mexico within a 5-year planning horizon, a Monte Carlo simulation model was utilized. Net returns were projected using simulated values for the distribution of input and product processes, establishing 2021 as base scenario. A stochastic modelling approach was employed to determine the economic and financial outlook. The findings reveal a panorama of economic and financial viability. Net income increased by 555%, return on assets rose from 3.36% in 2022 to 11.34% in 2026, and the probability of decapitalization dropped from 58% to 13%, respectively in the aforesaid periods. Similarly, the probability of obtaining negative net income decreased from 40% in 2022 to 18% in 2026. The technological, productive, and economic management of the production unit allowed for a favorable scenario within the planning horizon. There is a growing interest in predicting the economic sectors worth investing in and supporting, considering their economic and development performance. This research offers both methodological and scientific evidence to demonstrate the feasibility of establishing a planning schedule and validating the suitability of the pork sector for public investment and support.

## Introduction

Mexico is a country with a supply deficit in pork. According to statistical data, 2021 ended with an import volume (not including skin, brine, or other subproducts) of 969,358,505 kg of

**Funding:** -FEMC is awarded author - Grant number: 6498/2022CIB - Funder: Universidad Autónoma del Estado de México -URL:uaemex.mx - NO - The funders had no role in study design, data collection and analysis, decision to publish, or preparation of the manuscript.

**Competing interests:** The authors have declared that no competing interests exist.

pork [1], representing just over 12 million pigs [2]. While food production in Latin America mainly stems from small-scale farms [3], only 30% of these farms make it to a second generation, and only between 10% and 15% make it to a third generation [4]. In developing countries, the livestock sector encompasses almost one billion small-scale livestock production units that contribute with 40% of the agricultural GDP, and between 2% and more than 33% of the household income [5]. Given the complex scenario regarding generational transition in farms, methodical planning and economic and financial forecasting represent important tools for decision-making.

According to official data, medium-scale pig farming (between 81 and 1500 animals) contributes between 20% and 30% of the total pork production in Mexico. Since the opening of trade in the said country, pig farming has been one of the most affected productive sectors and has lost a large part of its production capacity [2,6,7]. The study of [8], concluded that a subsidy to the supply function of pork carcass would improve the benefit to society by increasing production and protecting consumers. Similarly, the latter authors evaluated the application of a 20% *ad valorem* tariff, concluding that the sector is sensitive to such tariff and the current policy protects national producers; however, it has a negative impact on both importers and consumers [9], in a market with a deficit in pork supply. According to [10], pork production is an economic sector that has a total multiplier of 3.40, which strategically positions it as a driving force of the economy. Therefore, it is crucial to assess whether to implement support and transfers to this sector.

A stochastic modelling approach of agricultural systems is a method used to evaluate agricultural management practices considering different risk factors, and to understand how such factors could affect the economic and financial viability, as well as the performance of the said systems in the future [11]. The stochastic model developed by [12], was used to aid vertically integrated pork companies in decision-making and coordination of production, taking into account the uncertainty associated to future price of pork. For their part, [13] used both Monte Carlo simulations and econometric models to assess the economic viability in the long term of Mexican pig farms considering four scenarios: Base scenario; hiring-in labor scenario; purchased inputs scenario: and zero subsidies scenario. Similarly, [14] evaluated the economic and financial viability of small-scale dairy systems in central Mexico. [15] conducted an analysis of competitiveness, profitability, and risk analysis in the cow-calf production system. While both the aforesaid studies demonstrated the importance of carrying out a prospective analysis that included the economic, financial, and political spheres, they were also useful for identifying improvement strategies in the productive and technical activities within the farms. The objective of this study was to evaluate the economic and financial performance of a Representative Pig Production Unit with an average inventory of 150 sows (RPPU150), located in a semi-tropical region of Mexico, considering a planning horizon of 2022–2026.

## Materials and methods

### RPPU150 description

The studied farm is located in a semi-tropical region of the State of Mexico, with an annual average temperature of 20.3°C, an altitude of 1,100 meters above sea level, a precipitation of 1,158 mm, and coordinates 18°59' and 19°14' north latitude, and 99°49' and 100°14' west longitude.

The production system is a farrow-to-finish operation (piglets that are reared to a weight of 110 kg). Pigs are housed in total confinement and separated according to physiological stage. The cycle begins in the breeding phase with insemination of the sows, once gestation is confirmed, pregnant sows are moved to gestation barns for 116 days, and one week prior to

farrowing are transferred to maternity barns for a 23-day lactation period. On average, 12.5 live piglets are born per litter, with a mortality rate of 16%, meaning that 10.5 piglets are weaned at the end of the phase.

Piglets are weaned with an average weight of 7.37 kg, then transferred to nursery barns to consume pre-starter feed for around 26 days until they reach an average weight of 19.28 kg. Subsequently, the developing pigs are moved to a starting area to receive a starter diet for 26 days until they reach an average weight of 35.45 kg. Ultimately, pigs are transferred to fattening barns for 78 days: during the initial 26 days, pigs are fed with a growing diet until they reach a weight of 55.78 kg. Over the subsequent 26 days, they are fed a developing diet until a weight of 80.27 kg is reached. Lastly, pigs are provided a finishing diet for another 26 days until they reach market weight of 110 kg. All economic and financial data were obtained directly from the evaluated farm accounting sheets. An approval for field site access was not required since no field tests were carried out. Only data provided by the pork producers was gathered.

### Ethics statement

The Ethics Committee of the Universidad Autónoma del Estado de México (México) approved the research protocol and information gathering tools. Prior to participation, the owner of the farm that conformed the analyzed RPPU150 was provided information about the overall aim and expected outcome of the present research. Due to the nature of the study, no animals were used to obtain data.

### The model

A Stochastic simulation approach was carried out to evaluate the economic and financial viability of a representative medium-scale pig farm, using an empiric distribution [16], used by [13,14] to evaluate Mexican pig farms and small-scale dairy systems, respectively. Historical data for the probability distributions were taken from the [17] database, which comprised 42 years of information (1981–2022). The studied model included data pertaining to the Mexican government to represent the national macroeconomic variables, while data from the International Monetary Fund (IMF) and the United States Department of Agriculture (USDA) were used for the international macroeconomic variables.

According to [16], Empirical multivariate probability distributions (MVE) were estimated for each variable of the analyzed production unit. Historical data from the past ten years were used to estimate the MVEs. The MVEs were then used to simulate stochastic projections of the variables for the next five years with 500 iterations using the software [18].

In order to estimate the parameters for a Multivariate Empirical (MVE) distribution, the following steps were conducted: 1) Random and non-random components estimation for each stochastic variable ($X\_it = \hat{a} + b*Trend + \hat{c}Z\_t$) were separated; 2) The random component of each stochastic variable ($\hat{e}\_it = X\_it - X\_it$). were calculated; 3) Residuals ($\hat{e}\_it$) were converted to relative deviates about their respective deterministic components; 4) The relative deviates ($D\_it = \hat{e}\_it/X\_it$) were sorted and pseudo-minimums and pseudo-maximums created for each random variable; 5) A probability of $P_{min} = Minimum\ S_{it}*1.000001$ to $P_{max} = maximum*S_{it}*1.000001$ with $S_{it} = Sorted[D_{it}\ from\ min\ to\ max]$ was assigned to each of the sorted deviates; 6) Calculate the M x M intra-temporal correlation matrix for the M random variables; and 7) The intertemporal correlation coefficients were calculated for the random variables. The seventh step completes the parameter estimation for a MVE distribution.

The complete MVE probability distribution was simulated with the Microsoft Excel add-in tool @RISK. The following steps were carried out: 1) A sample of Independent Standard Normal Deviates (ISND) was generated. ISND = Risknormal(0,1); 2) To simulate a MVE

distribution the ISNDs within each year of the simulation period (k = 1, 2,. . ., K) were correlated by multiplying the factored correlation matrix ($R_{ij}$) and eight of the values in the ISND vector (CSND_i = R_ij*ISND_i); 3) The inter-temporal correlation of the random variables was captured; 4) Equation Adjusted Correlated Standard Normal Deviates (ACSNDs) were transformed to Correlated Uniform Deviates (CUD). CUD_i = normsdist(ACSND_i); 5) $CUD_i$s were used to simulate random deviates for the empirical distribution of each variable $X_i$; and 6) The correlated fractional deviates were applied to their respective projected means making any needed adjustments for heteroscedasticity. $\tilde{X}_i = \hat{X}_i * (1 + CFD_i * E_i$, with correlated fractional deviates (CFD) and expansion factor ($E_i$).

The model projected economic and financial viability and provided information on trends for macroeconomic variables. In order to reduce estimation errors, trends in regional variables were estimated for the period of 2022–2026. The model operates annually at a strategic level and generates financial pro-forma reports that contain results used to calculate key output variables such as net incomes in cash, final cash reserves, change in net real capital, net present value (NPV) and rate of return over assets (RRA), which can inform decision making at both the farm and policy levels. The said financial reports are generated from functional equations that link pig production, sales, production and purchase of inputs, capital operations, and consumption and financing operations.

## Stochastic variables of the model

The inputs prices and yield of crops in the RPPU150 were obtained via meetings with producers and direct interviews. A Multivariate Empirical Distribution (MVE) was used to simulate the cost of crops and pigs. The probability distribution parameters were estimated with the Latin hypercube method, used as a sampling procedure for the simulation of pseudo-random numbers.

The use of such method ensures that the Coefficient of Variation (CV) and the mean for the simulated random variables are equal to the CV and the mean for the historic variables.

The stochastic values from the probability distributions are used in accounting equations to calculate production, receipts, costs, cash flows, and balance sheet variables for the project. Stochastic values sampled from the probability distributions make the financial statement variables stochastic, which are iterated 500 times using random values for the risky variables, the obtained information is used to estimate empirical probability distributions for the Key Output Variables (KOVs) such as present value of ending net worth, net present value, annual cash flows, amongst others. Providing forecasted KOVs that are used by decision makers [19,20].

## Economic and financial viability indicators

The analyzed indicators were total income (TI), total costs (TC), net income in cash (NIC), reserves in cash (RIC), NPV, RRA and cost to benefit ratio (C/B).

## Model assumptions

The analysis considered the following assumptions: (1) scale of production; (2) productivity; (3) farm infrastructure capacity and utilized; (4) technical coefficients are held constant during the planning horizon 2022–2026; (5) the technological level was also held unaltered; and (6) discount rate was established at 12%.

Economic values used were transformed in USD with a currency rate of 19.4143:1 USD: MXN according to Banco de Mexico (December 30, 2022).

**Table 1. Economic viability of a RPPU150 (Thousands USD).**

|  | **2021** | **2022** | **2023** | **2024** | **2025** | **2026** |
|---|---|---|---|---|---|---|
| Gross income | 869.05 | 911.74 | 960.46 | 1,015.28 | 1,058.45 | 1,111.82 |
| Total expenditure | 849.98 | 889.93 | 903.38 | 926.69 | 958.39 | 986.76 |
| Net income | 19.07 | 21.81 | 57.08 | 88.59 | 100.07 | 125.06 |
| Return on assets (%) | 3.36 | 1.50 | 6.59 | 10.21 | 10.24 | 11.34 |
| P(Cash Flow deficit) (%) | - | 40.00 | 34.00 | 24.80 | 24.00 | 18.00 |
| P(Refinancing) (%) | - | 20.60 | 17.80 | 10.80 | 6.00 | 4.60 |
| P(Return on assets < 0) (%) | - | 58.00 | 46.00 | 32.20 | 21.20 | 13.00 |

P = Probability.

## Results and discussion

Table 1 shows the economic viability with a noticeable increase in all accounting entries. Gross income grew by 27.93% within the planning horizon, while expenses had a 16.09% reduction, this allowed for a 36% average annual growth of net income. The probability of falling into negative indicators was considerably reduced, with the cash flow deficit dropping from 40% in 2022 to 18% in 2026; the probability of refinancing decreased by 16 percentage points, and the probability of a return on assets less than zero dropped from 58% in 2022 to 13% in 2026 Table 1. Economic viability of a RPPU150 (Thousands USD).

Net cash income performance within the planning horizon, according to the risk evaluation, showed that it is possible to fall in bankruptcy (5th percentile). However, 2026 economic panorama is a viable one (Fig 1).

The cost structure was mainly composed of feed (88.17% of total), followed by labor costs (5.79%), breeding costs (3.46%), and fuel costs (1.26%). Other concepts such as maintenance, veterinary service expenses, indirect expenses, transportation, and land costs, accounted for 0.79%, 0.47%, 0.03%, 0.02%, and 0.005% of the total costs, respectively.

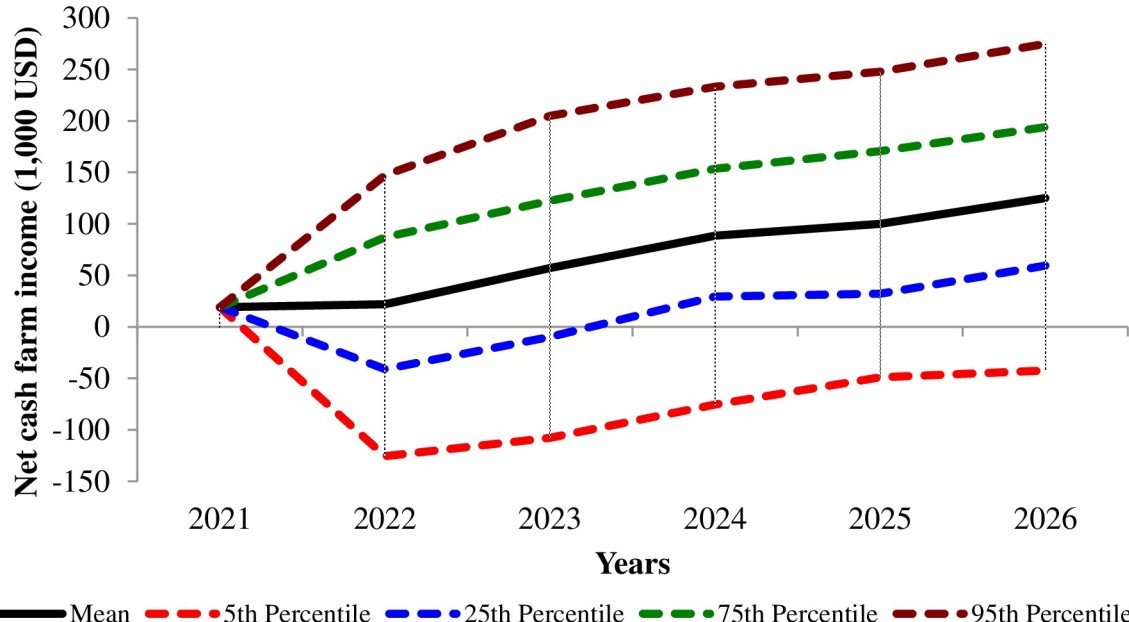

**Fig 1. Net cash farm income performance according to percentile and risk simulation.**

In the evaluated RPPU150, there is a null probability of obtaining a negative Net Present Value (NPV). The NPV in 2026 finished in 46.9964 thousand USD. The Internal Rate of Return (IRR) was 11.06%. The Cost/Benefit ratio was 0.98 and the Real Rate of Return was 0.07%.

Around 1.7 million tons (MT) of pork were produced in Mexico in 2021 [17], and the State of Mexico contributed with just over 22 thousand tons. Specifically, the said country had an average annual growth rate of 0.74, while the State of Mexico had a growth rate of -3.32 from 1980 to 2021.

In Mexico, the pig farming sector can be described in three historical and conjunctural periods [21]: an initial period of growth driven by the structural policies established in the second half of the 1970s; a second period of crisis in the sector that began in 1985 and ended in 1991; and a third period starting from 1991 with a growing trend until 2021, mainly driven by the consolidation of corporate pig farming systems.

The macroeconomic and sectoral policies adopted by Mexico in the last two decades has led to a change in the production structure of pig farming at the national and local level. While the State of Mexico is in the Central East region of Mexico, where growth and dynamism had positive effects during 1994 to 2012, the average annual growth rate for the said state was -1.31%.

One of the main effects observed in the Mexican pig industry was the impact of trade liberalization on the sector, mainly due to the competitive advantages of the United States over Mexico [22]. The difference between harm to producers and the benefit to consumers, namely $14.0 billion USD, remained in the hands of pork importers and traders, as in Mexico, among other causes, an asymmetric price transmission exists due to an oligopolistic market where retailers tend to quickly pass on price increases to consumers, while delaying the decrease in prices when they have decreased for the producer [23]. Such is the dependence on the external market that a 15% and 21% currency depreciation of the Mexican Peso would only imply an increase in national production of 0.3% and 0.4%, respectively [24].

The transition in the pig farming structure in the State of Mexico is mainly derived from technological progress, defined as feed conversion, and to a lesser extent, the price of feed and the price of pork, with average elasticity values of 1.174, -0.039, and -0.099, respectively [25].

In this regard, [26] determined that the best profit for pig farmers was obtained when they produced animals with a range in live weight between 66 and 162 kg. However, the production of heavier carcasses did not necessarily imply a higher profit [25]. The latter authors also found that the Technical Optimum Level (TOL) and Economic Optimum Level (EOL) for pork carcass weight were 94.47 kg and 90.96 kg, respectively, with profits of USD$ 162.1 and USD$ 173.0. In the case of secondary cuts, TOL and EOL were obtained with respective carcass weights of 85.4 kg and 85.3 kg, with profits of USD$ 236.7 and USD$ 236.9, while for tertiary cuts, 82.38 kg (TOL) and 82.26 kg (EOL) were equivalent to profits of USD$ 217.6 and USD$ 217.9.

The State of Mexico used to be one of the main pork producing states. Unlike national pork production, which recovered since 1991, the aforesaid state's pork industry was not able to do so, mainly due to the closure of a large part of its production units. While pork production at the national level grew an average annual rate of 2.63% from 1990 to 2021, it decreased by 1.20% in the State of Mexico. The observed trend in the production volume of pork in Mexico supports global projections that estimate an increase of pork primarily in low and middle-income countries [27], but the outlook is not the same for the pork industry in the State of Mexico. In this context, the estimation of financial projections has become a useful tool for assessing economic and financial viability of livestock systems. However, models must consider the conditions which prevail in the studied regions in order to reduce variability [28].

While the State of Mexico is a large consumer of pork, much of the said product is not produced within the state. One of the most significant changes from 2000 to 2019 was the concentration and commercialization of 21% of the slaughterhouses within the state. Additionally, 17 distribution centers and 24 slaughterhouses were located in the Central region of Mexico, representing 90% of the national traded volume [29].

Since 2006, a significant increase in the analysis of data pertaining to the pork sector has driven the development of models to assist decision-making [30]. The focus of these models is mainly related to diseases, DNA analysis, and feeding strategies.

Several studies have been conducted to measure the competitiveness of the pork industry at both the national and State levels in Mexico. By the end of the 2000s [6], reported favorable private profitability indicators in small, medium, and large-scale pig farms in the State of Mexico. Meanwhile [7], reported private gains in the sale of weaned piglets in small-scale pig farms (5 to 100 breeding sows). However, none of the latter studies included a risk analysis component like the present study.

In Brazil [31], applied a stochastic model in a typical full-cycle farm (1500 sows) to evaluate the impact on private and social profits of using locally produced co-products in finishing diets, finding a high coefficient of variation (75–87%) in profits due to volatility in prices and variations in pig biological parameters. These results demonstrate the importance of considering risk factors in strategic planning for livestock businesses.

On the other hand [32], generated a predictive model to evaluate the effect of making changes in the production system (increasing housing area to reach a higher market weight vs. feeding finishing feed at an earlier age), finding that the modification of feeding systems is the best alternative to increase farm income with a reduced investment. In our study, feeding represented 88% of the total cost structure.

The models of [26,32,33] allow for a clear visualization, in economic and biological terms, which option offers greater productivity. [34] reported that optimizing strategies related to feeding and distribution can increase gross income per pig by up to 10%. All the aforesaid models developed to estimate profitability did not consider micro and macroeconomic variables, and focused on biological components, as well as the price of some inputs for livestock feed.

The results of [35] warn that with the increase in the price of feed and land, the advantages of large-scale pig farms gradually become prominent, thus, forcing small and medium-scale farms to close. Additionally, since pig farming is inevitably linked with the environment and productivity levels, environmental legislation could have a strong impact on the future development of the pork sector [27].

[33] highlight the importance of individual farm modelling, since the combination of curves with different risk factors, production conditions, and payment curves in the same projection, could result in an underestimation of the economic and financial indicators.

Under the current technological, productive and management conditions, considering the macroeconomic and microeconomic outlook presented over the period 2022 to 2026, the RPPU150 showed favorable economic and financial indicators. While the studied scenario was favorable in its mean value, the results showed that there is a risk (5th percentile) of obtaining negative net cash income. Therefore, it is necessary to further analyze the farm's management of production and economic variables. A new prospective study is required given that the present analysis was conducted with a 12% interest rate, making the production unit highly vulnerable in financial terms. According to OECD estimations, a significant pressure on livestock input prices will predominate in the 2020–2030 decade [27].

Since in the analyzed pig production system, 88% of the cost structure was composed of feed, it is important to develop strategies in collaboration with regional agricultural producers

that generate benefits for both economic sectors. The goal is not only to ensure livestock feed inputs but also to create short marketing circuits that boost the local economy. While intensive pig production allows for the acquisition of a large volume of inputs, for a production unit with 150 sows (RPPU150), such economies of scale are not an option. Additionally, the acquisition of inputs from distant regions would have social and environmental implications [36].

While [37] demonstrated that small-scale pig farms (100 to 300 sows) were not competitive at private prices, profits at efficient prices increased in comparison to private prices, presenting a comparative advantage in international markets and contributing to 55% of the currency savings for the country that produced the pigs. However, it is important to consider the proper management of the macroeconomic policy related to the Mexican peso/US dollar exchange rate and interest rate since it may impose an additional tax per kilogram of live weight pig produced in Mexico. In this regard, [8] mentioned that a subsidy to the supply of pork would increase production, domestic consumption, and imports by 0.07%, 0.006%, and 0.003%, respectively. The Net Social Value would increase by 0.006%. Since Mexico is an importer of pork, achieving efficiency and competitiveness in this sector is crucial. [9] emphasize the role of possible import tariffs on pork as a protective measure for the sector. The latter authors report that a 20% tariff would increase domestic production by 0.4% (4,774 tons), reduce imports by 18.9% (19,141 tons), decrease consumption by 0.06% (14,367 tons), and reduce the Net Social Value by -0.1% (-$309,050,545 USD)

## Conclusions

The present results evidenced a consistent economic and financial viability scenario for the evaluated pig farm, under the current conditions, and within the analyzed planning horizon. Net income increased by 555%, return on assets rose from 3.36% in 2022 to 11.34% in 2026, and the probability of decapitalization dropped from 58% to 13%, respectively, in the aforesaid periods. Similarly, the probability of obtaining negative net income decreased from 40% in 2022 to 18% in 2026. The technological, productive, and economic management of the production unit allowed for a favorable scenario within the planning horizon. Even with the general favorable scenario, a risk (5th percentile) exists of falling into bankruptcy. It is important to determine the economic and financial performance within a planning horizon, since the analyzed system represents a typical small-scale pig farm in the State of Mexico, which represents 30% of the total pig inventory and 30% of pork production.

## Supporting information

**S1 Dataset.**
(XLSX)

**S2 Dataset.**
(XLSX)

## Acknowledgments

The authors wish to thank the pork producer that kindly provided the information for this study.

## Author Contributions

**Conceptualization:** Francisco Ernesto Martínez-Castañeda, Nicolás Callejas-Juárez.

**Data curation:** María Elena Trujillo-Ortega, Claudia Giovanna Peñuelas-Rivas, Elein Hernandez.

**Formal analysis:** Francisco Ernesto Martínez-Castañeda, Nicolás Callejas-Juárez.

**Funding acquisition:** Francisco Ernesto Martínez-Castañeda.

**Investigation:** Oscar Cuevas-Reyes, Elein Hernandez.

**Methodology:** Nicolás Callejas-Juárez.

**Project administration:** Francisco Ernesto Martínez-Castañeda.

**Resources:** Germán Gómez-Tenorio, Elein Hernandez.

**Validation:** Nicolás Callejas-Juárez.

**Writing – original draft:** Nathaniel Alec Rogers-Montoya.

**Writing – review & editing:** Nathaniel Alec Rogers-Montoya, María Elena Trujillo-Ortega.

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
