## [Decision Letter · Decision Letter 0]

27 Sep 2023

PONE-D-23-21718Economic and financial viability of a pig farm in central semi-tropical Mexico: 2022-2026 prospective

PLOS ONE

Dear Dr. Hernandez,

Thank you for submitting your manuscript to PLOS ONE. After careful consideration, we feel that it has merit but does not fully meet PLOS ONE’s publication criteria as it currently stands. Therefore, we invite you to submit a revised version of the manuscript that addresses the points raised during the review process.

Dear Authors

Reviewer/Reviewers reported that this study had serious flaws especially in construction of hypothesis, objective, plan of study and methodology. I, therefore, request you to revise the manuscript keeping in view the criticism of reviewer/reviewers. Failing to address the issues of manuscript will result in rejection. 

We look forward to receiving your revised manuscript.

Kind regards,

Aziz ur Rahman Muhammad

Academic Editor

PLOS ONE

“The authors wish to thank the pork producer that kindly provided the information for this study. and the research project 6498/2022CIB of the Universidad Autónoma del Estado de México for the financing of this research.”

“-FEMC is awarded author

- Grant number: 6498/2022CIB

- Funder: Universidad Autónoma del Estado de México

-URL:uaemex.mx

- NO -  The funders had no role in study design, data collection and analysis, decision to publish, or preparation of the manuscript.”

Additional Editor Comments:

Dear Authors

Reviewer/Reviewers reported that this study had serious flaws especially in construction of hypothesis, objectives, plan of study and methodology. I therefore request you to revise the manuscript keeping in view the criticism of reviewer/reviewers. Failing to address the issues of manuscript will result in rejection.

Reviewers' comments:

Reviewer's Responses to Questions

**Comments to the Author**

1. Is the manuscript technically sound, and do the data support the conclusions?

Reviewer #1: Partly

Reviewer #2: Yes

2. Has the statistical analysis been performed appropriately and rigorously? 

Reviewer #1: No

Reviewer #2: Yes

3. Have the authors made all data underlying the findings in their manuscript fully available?

Reviewer #1: No

Reviewer #2: Yes

4. Is the manuscript presented in an intelligible fashion and written in standard English?

Reviewer #1: Yes

Reviewer #2: No

5. Review Comments to the Author

Reviewer #1: The authors used a model to estimate a pig farm's economic and financial viability in central sub-tropical Mexico within a 5-year planning horizon. Also, they applied a Monte Carlo simulation.

The authors mentioned several similar studies. However, they also stated that these studies did not include a risk analysis component like the current study. I am not confident this is enough to justify the novelty of this paper. Risk analysis in investment analysis has various applications in the extant literature.

My main concerns are about the methodology. The authors should revise this section. All steps and procedures have to be carefully explained regarding the potential reproducibility. This is primordial to increase the visibility of the research.

My main concerns:

MATERIALS AND METHODS

- Please choose where appropriate and describe the package(s) you used to perform your quantitative analysis.

The Model:

- The authors state they used the probability distributions of Zavala-Pineda et al. (2012) and Posadas-Domínguez et al. Is that right? The authors must clarify which probability distributions they ( triangular, Gaussian, etc....) used and why they were adequate for the case study;

- It is not clear what exactly the authors are analysing. I would expect a representative cash flow (a figure) representing all variables considered;

- What are the necessary investments and costs for the studied farm? What are the empirical data applicable to this case study?

- In summary, this section lacks details to provide the readers with all the information to understand (if necessary) to replicate the study in different contexts.

Stochastic Variables of the Model:

- Again, the authors provide a too-short description of the methodological steps. How many simulations were performed? Please provide details and clarify all steps.

Results And Discussion:

- What are the contributions and the practical implications of this study? I expect a profound discussion on this matter in this section.

Minor revisions:

- Please see the attached file.

Reviewer #2: Abstract

1. A short background description is required, and the research objectives must be clear.

2. The research method must be clear, accompanied by the number of samples.

3. Findings are sorted according to objective research.

Introduction

1. The opening sentence of the first paragraph of the introduction should be a general statement without using citations.

2. The introduction must explain the importance of the research based on previous research references.

Materials and Methods

1. Create it in a table, then describe it.

Results and Discussion

1. Add discussion to Table 1 that is supported by previous research.

2. Add discussion to Figure 1 that is compared with previous research.

Comments

Finally, I recommend the author make major revisions to the manuscript.

6. PLOS authors have the option to publish the peer review history of their article (what does this mean?). If published, this will include your full peer review and any attached files.

Reviewer #1: No

Reviewer #2: No

---

## [Author Response · Author response to Decision Letter 0]

10 Nov 2023

According to reviewers’ suggestions the following changes were made to the original manuscript:

Title page:

Lines 3 to 5: Ampersand symbol was included.

Line 24: The statement “&These authors also contributed equally to this work” was included.

Introduction:

Line 41: number “1” was replaced by the word “one”.

Material and Methods:

Line 96 to 98: The following statement was included “An approval for field site access was not required since no field tests were carried out. Only data provided by the pork producers was gathered.”

Line 143: Table 1 is cell-based.

Regarding the reviewer question about the model used: “The authors state they used the probability distributions of Zavala-Pineda et al. (2012) and Posadas-Domínguez et al. The authors must clarify which probability distributions (triangular, Gaussian, etc....) they used and why they were adequate for the case study”

Indeed, the probability distributions of Zavala-Pineda et al. and Posadas-Domínguez et al. were used. However, it should be clarified that both latter studies employed an empirical distribution based on the model developed by Richardson, J.W., S.L. Klose, and A.W. Gray. 2000. An applied procedure for estimating and simulating multivariate empirical (MVE) probability distributions in farm level risk assessment and policy analysis. J. Agric. Appl. Econ. 32(2):299-315.

The following paragraph was included:

Line 110-113: According to [16], Empirical multivariate… …the next five years with 500 iterations using the software [18].

Regarding the reviewer question about the package used to perform the quantitative analysis:

The package used to make the estimations was SIMETAR©

Regarding the reviewer question: “It is not clear what exactly the authors are analyzing. I would expect a representative cash flow (a figure) representing all variables considered”

While a figure representing all considered variables was not included in the manuscript, the following paragraphs were included for a better understanding of the Materials and Methods section:

Line 114 to 124: The following paragraph was included “In order to estimate the parameters for a Multivariate Empircal (MVE) distribution… …. The seventh step completes the parameter estimation for a MVE distribution.”

Line 125 to 135: The following paragraph was included “The complete MVE probability distribution was simulated… … fractional deviates (CFD) and expansion factor (Ei).” 

Regarding the reviewer question: “What are the necessary investments and costs for the studied farm? What are the empirical data applicable to this case study?”

The following statement was included: 

Line 93 to 94: “All economic and financial… …accounting sheets.”

Regarding the reviewer observation about the Stochastic Variables of the Model: “Again, the authors provide a too-short description of the methodological steps. How many simulations were performed? Please provide details and clarify all steps.”

The following paragraph was included:

Line 151 to 157: “The stochastic values from the probability distributions… … KOVs that are used by decision makers [19, 20].”

Results and Discussion

Regarding the reviewer question: “What are the contributions and the practical implications of this study? I expect a profound discussion on this matter in this section.”

The following paragraphs were included:

Line 273 to 279: “Since in the analyzed pig production system… … regions would have social and environmental implications [36].

Line 280 to 292: “While [37] demonstrated that small-scale pig farms… … Net Social Value by -0.1% (-$309,050,545 USD)”

Acknowledgements

Line 254: The statement “The authors wish to thank the pork producer that kindly provided the information for this study, and the research project 6498/2022CIB of the Universidad Autónoma del Estado de México for the financing of this research” was replaced by “The authors wish to thank the pork producer that kindly provided the information for this study”.

References

Following the journal’s submission guidelines, the citing style was changed to numbers in square brackets (e.g. [1], [2]) and references are listed in order of appearance. 

The following citations and references are included in the manuscript to support the modifications made in the Materials and Methods section and in the discussion:

16. Richardson et al. (2000); 17. Richardson et al. (2006); 18. Richardson et al. (2007); 36. Villavicencio-Guitíerrez et al. (2021); and 37. Barrón-Aguilar et al. (1995).

---

## [Decision Letter · Decision Letter 1]

20 Nov 2023

PONE-D-23-21718R1Economic and financial viability of a pig farm in central semi-tropical Mexico: 2022-2026 prospectivePLOS ONE

Dear Dr. Hernandez,

Thank you for submitting your manuscript to PLOS ONE. After careful consideration, we feel that it has merit but does not fully meet PLOS ONE’s publication criteria as it currently stands. Therefore, we invite you to submit a revised version of the manuscript that addresses the points raised during the review process.

Dear AuhtorsYou have not fully addressed the issues raised by one of the reviewer. The other reviewer is busy and did not responded yet. I would like to invite another reviewer after this minor revision so that quality of Journal should be insured.

We look forward to receiving your revised manuscript.

Kind regards,

Aziz ur Rahman Muhammad

Academic Editor

PLOS ONE

Journal Requirements:

Additional Editor Comments:

Dear Authors

you have not fully addressed the issues raised by one of the reviewer. The other reviewer is busy and did not responded yet. I would like to invite another reviewer after this minor revision so that quality of Journal should be insured.

Reviewers' comments:

Reviewer's Responses to Questions

**Comments to the Author**

1. If the authors have adequately addressed your comments raised in a previous round of review and you feel that this manuscript is now acceptable for publication, you may indicate that here to bypass the “Comments to the Author” section, enter your conflict of interest statement in the “Confidential to Editor” section, and submit your "Accept" recommendation.

Reviewer #1: All comments have been addressed

2. Is the manuscript technically sound, and do the data support the conclusions?

Reviewer #1: Yes

3. Has the statistical analysis been performed appropriately and rigorously? 

Reviewer #1: Yes

4. Have the authors made all data underlying the findings in their manuscript fully available?

Reviewer #1: No

5. Is the manuscript presented in an intelligible fashion and written in standard English?

Reviewer #1: Yes

6. Review Comments to the Author

Reviewer #1: Dear authors:

I would expect a response letter as a table with each concern justified and/or addressed. I am not sure the way I got the response is poorly organized maybe because of the journal platform. But it is ok, I was able to see your response.

Some recommendations I made were not attended. However, in a general perspective the author´s response is acceptable.

Also, in this version there are still revisions regarding grammar. I marked one in the attached file. I strongly recommend a careful revision.

7. PLOS authors have the option to publish the peer review history of their article (what does this mean?). If published, this will include your full peer review and any attached files.

Reviewer #1: No

---

## [Author Response · Author response to Decision Letter 1]

3 Jan 2024

Dear editor, 

The comments have been responded and a revised version has been submitted with the rebuttal letter as requested. The letter includes a detailed response o the comments. Overall, the main comments have been responded: 

1. The list of references was reviewed, no citations were removed; five citations and respective references were included in the manuscript to support the modifications made in the first revision. 

2. The grammar concerns have been revised. 

3. We believe we have addressed all the comments made by the reviewer; however, we did not present the variables as a cash flow because we considered that the number of variables involved would result in a too large figure. In the first revision, we preferred to add additional paragraphs to the Materials and Methods section to clarify how the parameters used to make the estimations were obtained.

4. We apologize but we are not clear on what is being requested. We are sharing an Excel file that contains the input data and results obtained (output data from the software). We don't have our database available in a repository; however, we are willing to share the database upon reasonable request. 

Thank you. 

Kind regards.

---

## [Decision Letter · Decision Letter 2]

1 Feb 2024

Economic and financial viability of a pig farm in central semi-tropical Mexico: 2022-2026 prospective

PONE-D-23-21718R2

Dear Dr. Hernandez,

We’re pleased to inform you that your manuscript has been judged scientifically suitable for publication and will be formally accepted for publication once it meets all outstanding technical requirements.

Kind regards,

Aziz ur Rahman Muhammad

Academic Editor

PLOS ONE

Additional Editor Comments (optional):

Dear Authors

Thanks for revising the manuscript according to suggestions of reviewers. good luck

Reviewers' comments:

Reviewer's Responses to Questions

**Comments to the Author**

1. If the authors have adequately addressed your comments raised in a previous round of review and you feel that this manuscript is now acceptable for publication, you may indicate that here to bypass the “Comments to the Author” section, enter your conflict of interest statement in the “Confidential to Editor” section, and submit your "Accept" recommendation.

Reviewer #1: All comments have been addressed

2. Is the manuscript technically sound, and do the data support the conclusions?

Reviewer #1: Yes

3. Has the statistical analysis been performed appropriately and rigorously? 

Reviewer #1: Yes

4. Have the authors made all data underlying the findings in their manuscript fully available?

Reviewer #1: Yes

5. Is the manuscript presented in an intelligible fashion and written in standard English?

Reviewer #1: Yes

6. Review Comments to the Author

Reviewer #1: Dear authors:

To save time I suggest in future revisions (other submissions) always providing a table and each revision treated separately.

For now, I am happy with your efforts on improving this paper.

7. PLOS authors have the option to publish the peer review history of their article (what does this mean?). If published, this will include your full peer review and any attached files.

Reviewer #1: No

---

## [Editor Report · Acceptance letter]

19 Mar 2024

PONE-D-23-21718R2 

PLOS ONE

Dear Dr. Hernandez, 

I'm pleased to inform you that your manuscript has been deemed suitable for publication in PLOS ONE. Congratulations! Your manuscript is now being handed over to our production team.

Kind regards, 

on behalf of

Dr. Aziz ur Rahman Muhammad 

Academic Editor

PLOS ONE